# Large Prostate Volume Does Not Negatively Impact Health-Related Quality of Life in Patients with Prostate Cancer Treated with Ultrahypofractionated Stereotactic Body Radiotherapy

**DOI:** 10.3390/jpm13020233

**Published:** 2023-01-28

**Authors:** Piotr Milecki, Anna Adamska, Anna Rucinska, Grzegorz Pałucki, Agnieszka Szumiło, Agnieszka Skrobała, Agata Jodda, Michał Michalak

**Affiliations:** 11st Radiotherapy Department, Greater Poland Cancer Centre, 61-866 Poznań, Poland; 2Chair of Electroradiology, Faculty of Medicine, Poznan University of Medical Sciences, 60-780 Poznań, Poland; 3Department of Medical Physics, Greater Poland Cancer Centre, 61-866 Poznań, Poland; 4Department of Statistics, Faculty of Medicine, Poznan University of Medical Sciences, 60-780 Poznań, Poland

**Keywords:** prostate cancer, prostate volume, quality of life, stereotactic radiotherapy, cyberknife

## Abstract

Background: Survival outcomes after primary radiotherapy for localized prostate cancer (PCa) are excellent, regardless of the specific treatment modality. For this reason, health-related quality of life (HRQOL) has come to play an ever more important role in treatment selection. Stereotactic body radiation therapy (SBRT) is increasingly used to treat patients with PCa. However, the impact of prostate volume on HRQOL is not clear. In this study, we aimed to determine whether a large prostate volume negatively influences HRQOL outcomes in patients undergoing ultrahypofractionated SBRT. Material and Methods: We conducted a prospective study of 530 men with low- and intermediate-risk localized PCa. All patients were treated from 2013 to 2017 with SBRT (Cyberknife system). HRQOL data were collected at baseline (pre-treatment), immediately after treatment, and at 12 and 24 months. QOL variables were assessed with the European Organization for Research and Treatment of Cancer QLQ-C30 and PR-25 module. Differences in the QLQ-C30 scales were considered clinically relevant when the change was >10 points. For the analysis, patients were classified into two groups according to prostate volume (≤60 vs. >60 cm^3^). Results: The prostate volume was ≤60 cm^3^ in 415 patients (78.3%) and >60 cm^3^ in 115 (21.7%). No between-group differences were observed at baseline for any of the following variables: clinical stage; hormonal therapy; marital status; educational level; or employment status. No clinically-significant deterioration (functional and symptom scales) was observed in either group between the baseline and 24-month assessment. There were no clinically-relevant differences between the groups on any of the HRQOL variables, regardless of the prostate volume. Conclusions: This study shows that a large prostate volume (>60 cm^3^) does not appear to negatively impact HRQOL outcomes at two years in patients with localized prostate cancer treated with ultrahypofractionated SBRT administered with the CyberKnife system.

## 1. Introduction 

In recent decades, the development of advanced radiotherapy techniques such as stereotactic body radiation therapy (SBRT) has enabled the administration of ultrahypofractionated radiation doses, thus leading to better treatment outcomes in patients with prostate cancer (PCa) [1]. Importantly, the safety and effectiveness of SBRT for the treatment of localized PCa are supported by a large body of evidence [2,3].

Given the excellent survival outcomes achieved with SBRT, health-related quality of life (HRQOL) has become an increasingly important factor in selecting the optimal treatment approach. Prostate gland volumes can vary widely in patients with PCa, although the volume is usually less than 60 cm^3^ in most cases. However, due to the presence of benign prostate hyperplasia—particularly in older men—a substantial proportion of patients have a prostate gland that is >60 cm^3^. The presence of a large prostate is important given the reported association between large prostate glands and higher rates of genitourinary (GU) and gastrointestinal (GI) toxicity after radiotherapy, which could negatively influence HRQOL outcomes [4,5]. Nevertheless, only a few studies have been performed to determine the impact of prostate volume on HRQOL in patients treated with SBRT, with equivocal results [6,7]. In short, due to the lack of robust data, the influence of prostate gland volume on HRQOL after SBRT is not well-understood.

In this context, we conducted the present prospective study to determine the effect of pre-treatment prostate volume on HRQOL outcomes up to 24 months after treatment in patients with clinically-localized PCa treated with ultrahypofractionated SBRT.

## 2. Materials and Methods

### 2.1. Patients

This was a prospective trial (CyberProst trial) registered with the National Institute of Health U.S. National Library of Medicine registry (ClinicalTrials.gov identifier: NCT 03225235). The trial was approved by the institutional bioethics review board at Poznan University of Medical Sciences (Poland).

All patients included in this study underwent SBRT between September 2013 and May 2017 at the Greater Poland Cancer Center (GPCC). The inclusion criteria were: histologically-confirmed low or intermediate-risk PCa, defined as follows: stage T1c to T2a-c; Gleason score < 8; prostate-specific antigen (PSA) < 20 ng/mL; no evidence of nodal involvement or distant metastases; and World Health Organisation (WHO) performance status 0–1. Written informed consent was required prior to study enrolment. All patients were treated with SBRT. Clinical stage was determined according to the 6th edition of the American Joint Committee on Cancer. Exclusion criteria were: <2 years of follow-up; clinical involvement of lymph nodes; distant metastases on pre-treatment imaging; prior PCa-directed locoregional therapy and/or prior pelvic irradiation.

To evaluate the impact of prostate volume on HRQOL, the patients were divided into two groups according to prostate volume: ≤60 cm^3^ and >60 cm^3^.

### 2.2. Radiotherapy

For treatment planning purposes, four gold fiducial markers were inserted into the prostate gland. One week later, a computed tomography (CT) scan was performed (slice thickness, 2 mm) followed by T2-weighted magnetic resonance imaging (MRI). The CT and MRI images were then fused for treatment planning. The clinical target volume (CTV) was defined by the prostate capsule and the proximal seminal vesicles. The planning target volume (PTV) was created by expanding the CTV margins by 3 mm posteriorly and 5 mm in all other directions. The following structures were contoured as organs at risk (OAR): rectum, bladder, penile bulb, and both femurs. The prostatic urethra was not included as an OAR because the prescription isodose line was limited to ≥75% to restrict the maximum dose to 125% of the prescription dose. SBRT was delivered to patients using the CyberKnife robotic radiosurgery system (Accuray Inc., Sunnyvale, CA, USA). Fiducial-based tracking was used to account for inter- and intra-fraction prostate motion.

A total dose of 35 or 36.25 Gy to the PTV was delivered in five fractions of 7.0 or 7.25 Gy. Before simulation (planning CT and MRI) and prior to each SBRT fraction, patients were instructed to drink approximately 500 mL of water 20–30 min before the session to fill their bladder to a comfortable level. They were also instructed to present with an empty rectum.

### 2.3. HRQOL Assessment and Follow-Up

All patients were seen at the GPCC outpatient department one month after completion of radiotherapy and then every three months thereafter for two years. All patients were asked to complete QOL questionnaires at baseline, immediately after completion of radiotherapy treatment, and at 12 and 24 months. HRQOL was assessed prospectively using the validated European Organization for Research and Treatment of Cancer (EORTC) QLQ-C30 (v. 3.0). The EORTC QLQ-C30 consists of 30 items to assess functioning (physical, role, emotional, social, and cognitive), and symptoms (fatigue, nausea and vomiting, pain, dyspnoea, insomnia, appetite loss, constipation, diarrhoea and financial difficulties) and one item to assess global health status (QoL scale) [8]. Differences in the QLQ-C30 scales at the different time points were considered clinically relevant when the change was >10 points, following the approach used by other authors [9]. We also administered the QLQ-PR25, a specific module from the EORTC, which consists of 25 items to assess six domains (urinary symptoms, incontinence aid, bowel symptoms, hormonal treatment-related symptoms, sexual activity, and sexual functioning) [10].

The domain scores of the QLQ-C30 and PR25 modules were calculated according to the scoring manual provided by the EORTC QOL group as follows: raw scores were linearly transformed to a scale of 0 to 100, with 100 representing the worst symptom status or the best functional status; scale scores were not computed when >50% of the responses of the corresponding scale were missing values [11,12].

For the EORTC QLQ-C30, higher scores on the functional scales and on the general health status item indicate better QOL, whereas higher scores on the symptomatic scales indicate greater symptom severity. For the EORTC QLQ-PR25, higher scores on the functional scales indicate a better functioning; by contrast, higher scores on the symptomatic scales indicate greater symptom severity [10]. The QLQ-C30 and PR25 questionnaires were scored according to published recommendations [8,13].

### 2.4. Statistical Analysis

The results of the EORTC QLQ-30 and QLQ-PR25 domains are presented as means with standard deviation (SD). The domains were compared at four time points (baseline, immediate post-SBRT, and at 12 and 24 months after SBRT) using repeated measures analysis of variance (repeated measures ANOVA). Tukey’s test was used for the post-hoc analysis of specific contrasts Student’s T test was used to compare differences between the baseline and 24-month assessments. Categorical data were compared by the chi-square test for independence. For tumor stage categories, the comparison was performed by a test for proportion. The results are presented as numbers with percentages. The statistical analyses were performed with the statistical package STATISTICA, V.13 (TIBCO Software Inc., Palo Alto, Santa Clara, CA, USA). All tests were considered significant at *p* < 0.05.

## 3. Results

A total of 530 patients met all inclusion criteria and were included in the study. The prostate volume was ≤60 cm^3^ in 415 patients (78.3%) and >60 cm^3^ in 115 (21.7%). The mean prostate volume for the cohort was 42.9 cm^3^ (range: 29.2–158.9). According to National Cancer Comprehensive Network (NCCN) classification criteria, 254 patients (48%) were considered low-risk and 276 (52%) intermediate-risk. Table 1 summarizes the patients’ demographic data (age, educational level, and marital status), clinical stage, and treatment details according to prostate volume at baseline.

As Table 1 shows, there were no significant between-group differences in clinical stage, hormonal therapy, marital status, educational level, or employment status. However, significant differences were observed in mean age, Gleason score, and total SBRT dose. Patients in the smaller prostate volume group were slightly younger (68 vs. 70 years, respectively; *p* = 0.0002). A significantly higher proportion of patients in the large prostate volume group (22.8% vs. 9.9%) were treated with 35 Gy vs. 36.25 Gy of SBRT due to difficulties in meeting dose constraint criteria for OARs in patients with large prostate volumes. Nevertheless, dose reduction from 36.25 to 35 Gy is the standard approach in these cases and has been shown to have no negative impact on oncological efficacy.

Table 2 shows the EORTC C-30 and PR-25 scores at the four time points (baseline, completion of SBRT, 12- and 24-months post-treatment) for the two groups.

As Table 2 shows, global health status declined in both groups immediately after treatment finalisation. However, over time, health status gradually recovered, returning to the baseline level by month 24. A similar pattern was observed in both groups, with no statistically significant between-group differences observed for most of the EORTC QLQ C30 and PR-25 items. However, differences were observed in the following variables: social functioning, pain, insomnia, financial difficulties, bowel symptoms, and sexual activity. Nonetheless, given that the post-treatment decreases in the scores on these items was <5 points in all cases, the decline was not considered clinically relevant. 

In terms of HRQOL in the large prostate volume group, treatment had a statistically significant (but not clinically relevant) negative impact on several variables. By contrast, the impact on sexual activity was considered clinical meaningful, with an increase (worsening) in mean scores from 63.2 points at baseline to 70.8 points at month 24 (Table 2). For the other items on the functional and symptom scales, no statistically or clinically significant deterioration was observed in either group. 

Table 3 compares changes from baseline to month 24 on the QLQ-C30 and PR25 items in both groups. As that table shows, significant between-group differences were observed for several variables, indicating a greater deterioration in the large prostate volume group for pain, insomnia, appetite loss, financial difficulties, bowel symptoms, and sexual activity. Nevertheless, these differences between groups were not clinically relevant. 

## 4. Discussion

The present prospective study was conducted to determine the impact of a large prostate volume (>60 cm^3^) on HRQOL-related treatment outcomes in patients undergoing ultrahypofractionated SBRT for low- or intermediate-risk PCa. Our data show that the presence of a larger prostate volume did not have a clinically-significant negative impact on HRQOL in this patient population at 24 months post-treatment.

Survival outcomes in patients with favourable-risk PCa are good, regardless of the specific treatment modality, which explains why QOL has come to play an increasingly important role in treatment selection. In recent years, interest in treatment modalities such as SBRT that can shorten the course of radiotherapy from weeks to only a few days has increased substantially. Several factors underlie this interest in shortening treatment times, including greater convenience for the patient, lower treatment costs, and the potential clinical advantages of delivering a higher total radiobiological dose per treatment than can be achieved with conventional radiotherapy. Several clinical trials are currently underway to compare conventional long-course radiotherapy to ultrahypofractionated radiotherapy (up to 2 weeks), including the following PACE (NCT01584258) trial [14]. HRQOL outcomes will be assessed in all of these trials. Physician-reported late grade 3–5 GI and GU toxicity rates with ultrahypofractionated SBRT range from 0–1% and 0–2.5%, respectively [15,16,17]. However, because patients in routine clinical practice tend to be older than those enrolled in clinical trials, the proportion of patients in real-life practice with a large prostate gland is higher, despite the use of short course antiandrogen deprivation therapy.

At present, the optimal radiation treatment modality for patients with large prostate glands remains unclear, mainly due to the lack of robust data from large cohorts. In this regard, we conducted this study to help fill this knowledge gap by comparing HRQOL outcomes according to prostate volume. Our findings show a temporary worsening in several QoL domains (pain and urinary symptoms, among others) at completion of SBRT, but over time HRQOL improved, with no clinically-relevant differences between baseline and the 24-month assessment (Table 2).

The cut-off volume of 60 cc for prostate gland volume analysis was chosen because of two main reasons: The first is that many papers which deal with brachytherapy as a treatment for prostate cancer use exactly this level of volume as a threshold for less versus a higher level of side effects related to treatment. So, because SBRT could be considered a kind of competition to brachytherapy, regarding which method of therapy is more suitable for our patient, information regarding this parameter could impact the choice of therapy methods. 

In the current randomized clinical trial (GU 005) comparing SBRT to IMRT, according to the inclusion criteria, patients are accepted into the study with a prostate volume of less than 60 cc.

The second reason is practical too, because prostate cancer in many cases can coexist with a non-malignant enlarged prostate gland and the practical volume cut-off for surgeons (urologists) who operate using different methods for a non-malignant (Benign Prostate Hyperplasia or BPH) or a malignant gland is a volume of about 60 cc. In general, a prostate gland with a higher volume could be an issue for radiation oncologists as well urologists. From a radiotherapy point of view, in the case of a prostate with a larger volume, it is harder to protect the Neurovascular Bundles (NVBs) which are located very closely to the capsule of prostate. That is one reason why radiotherapy in such a situation is more challenging.

Some previous studies suggest that the adverse effects of radiotherapy could be greater in patients with larger prostate glands, which in turn would negatively impact HRQOL [4,18]. However, the presence of adverse effects does not necessarily imply worse HRQOL. Chen et al. prospectively evaluated the influence of SBRT on urinary incontinence in patients with PCa [19]. Those authors evaluated QOL at 3 years post-retreatment in 204 patients treated with SBRT (35–36.25 Gy in five fractions), finding a significant decrease in the Expanded Prostate Cancer Index Composite (EPIC) urinary incontinence score. Although the decline was not clinically significant, the authors did observe an association between prostate volume and urinary incontinence scores. Woo et al. evaluated 515 patients treated with SBRT (35–36.25 Gy in five fractions). Prostate volume data were available for 336 patients. In this group, the incidence of grade two and three urinary toxicity was higher in patients with a large prostate volume (>60 cm^3^), with a trend towards statistical significance [20]. Conversely, another study evaluated 216 patients treated with SBRT (35–36.25 Gy in five fractions), finding no correlation between prostate volume and urinary symptoms at 2 years [21]. However, it is important to emphasise that the mean prostate volumes in those studies were not large: in the studies by Woo et al. and Chen et al., the mean volumes were 39 and 65.3 cm^3^, respectively.

Janowski et al. retrospectively reviewed 57 patients with a median prostate volume of 63 cm^3^ (versus 42.9 cm^3^ in our study) treated with 35 to 36.25 Gy of SBRT (five fractions) [22]. Grade three urinary tract toxicity was observed in only two patients (3.5%), but the authors were unable to draw any conclusions on the impact on HRQOL due to limited data on toxicity, and especially HRQOL data in patients with large prostates (>60 cm^3^).

Overall, we found that changes in the domains of the core EORTC QLQ-C30 had only a minor effect on HRQOL in the two groups. The 4–10% decline in global health status from baseline to the 24-month evaluation represents only a minor change in HRQOL. Similarly, the decline observed in physical and role functioning in the functional domains was also minimal. Of the nine symptom domains, only fatigue, insomnia, and diarrhea showed a statistically significant change, and even these changes were mainly of minimal clinical significance (although the increase in fatigue and insomnia indicates a moderate decrease in QOL).

In general, the EORTC QOL results interpretation for daily clinical practice is based on the manual provided by the EORTC. According to the manual the raw QLQ-C30 scores can be transformed to scores ranging from 0 to 100. The use of these transformed scores has several advantages, but transformed scores may be difficult to interpret. For example, what does an emotional function score of 60 or a difference of 15 mean? Also, there are no grounds for regarding an emotional function score of 60 as being equally good or bad as scores of 60 on the other functioning scales. However, there are a number of ways to ease the interpretation of QLQ-C30 results.

Current approaches to defining clinical thresholds for the EORTC QLQ-C30 vary substantially and each has its own strengths and weaknesses. The simplest and most straight-forward approach is to rely on the wording of the item or response categories themselves, and classify a patient as having a clinically important problem if s/he responds with at least “a little” for any given item or domain assessed on the four point response scale (i.e., “not at all”, “a little”, “quite a bit”, and “very much”). This approach can be problematic as it uses the same threshold for all QLQ-C30 domains. The QLQ-C30 items differ in item difficulty (i.e., they assess different levels of severity of a problem or symptom), suggesting that scores are not directly comparable across domains. For example, reporting “quite a bit” of vomiting probably indicates a different symptom level than reporting “quite a bit” of trouble with sleeping. So, currently we don’t have the ideal toll for interpretation of results for daily practice. The proposition of a cut-off of 10 points as clinical relevance is the most used in the interpretation of results from different studies, but in addition it should be included with the raw statistical results.

### Strengths and Limitations

This study has several limitations. One limitation is the lack of randomisation. Another limitation is the sequential treatment of the patients, which might have at least partially influenced HRQOL outcomes. In addition, we only assessed HRQOL up to two years post-treatment. HRQOL outcomes could change over time, which is why long-term data are needed to confirm whether the HRQOL findings are maintained over time. However, many studies have demonstrated little HRQOL change after two years for conventional radiotherapy and some studies have shown the same for SBRT, with one study demonstrating only small variations in urinary and bowel QOL (EPIC questionnaire) even up to eight years after treatment [23,24]. The main strengths of this study include the prospective study design (with baseline measurements and validated QOL questionnaires) and a homogenous population-based cohort with similar demographic characteristics and radiotherapy parameters (dose, technique, etc.). In addition, response rates to most items on the questionnaires were high, except for a few questions on sexual function.

## 5. Conclusions

The findings of this prospective study show that a large prostate volume does not appear to negatively influence HRQOL outcomes in patients with favourable-risk prostate cancer treated with SBRT. These results suggest that patients with a prostate volume >60 cm^3^ can be safely treated with high dose SBRT with minimal impact on HRQOL, as evidenced by the minimal between-group differences observed in this study in gastrointestinal and urinary toxicity rates and on most EORTC QOL items. This study adds valuable data to the body of evidence supporting the value of SBRT for the treatment of low- and intermediate-risk prostate cancer, indicating that SBRT is a suitable treatment option for patients with larger prostate volumes.

## Figures and Tables

**Table 1 jpm-13-00233-t001:** Demographic, clinical, and treatment characteristics of the study population by prostate volume.

Variable	Prostate Volume ≤ 60 cm^3^ (n = 415)	Prostate Volume > 60 cm^3^ (n = 115)	Total	*p* Value
Mean age (range) at baseline, years	68 (51–80)	70 (53–80)	530	0.0002
Tumor stage (T category):				
T1	298 (72%)	86 (74.8%)	384	0.5376
T2a	27 (6.5%)	8 (7%)	35	0.8784
T2b	60 (14.5%)	12 (10.4%)	72	0.2566
T2c	30 (7.0%)	9 (7.8%)	39	0.7687
Pre-treatment PSA:				
<10 ng/mL	252 (60.1%)	74 (64.4%)	326	0.4043
10–20 ng/mL	163 (39.9%)	41 (35.7%)	204	
Gleason score:				
2–6	285 (68.8%)	90 (78.3%)	375	0.0475
7	130 (31.2%)	25 (21.7%)	155	
Radiotherapy dose:				
35.0 Gy	41 (9.9%)	26 (22.8%)	67	0.0002
36.25 Gy	374 (90.1%)	89 (77.2%)	463	
Comorbidity:				
Yes (any)	356 (85.9%)	98 (85.6%)	454	0.9177
No	59 (14.1%)	17 (14.4%)	69	
Educational level:				
Less than High School	161 (38.9%)	38 (33.3%)	199	0.4768
High School and above	254 (61.1%)	77 (66.7%)	331	
Employment status:				
Employed	42 (10.2%)	10 (9.4%)	52	0.7865
Unemployed	373 (89.8%)	97 (90.6%)	470	
Marital status:				
Married	90 (21.8%)	11 (10.2%)	101	0.0632
Unmarried	325 (78.2%)	104 (89.8%)	429	
Hormonal treatment at baseline:				
Yes	187 (45.1%)	43 (37.4%)	230	0.142
No	228 (54.9%)	72 (62.6%)	300	

**Table 2 jpm-13-00233-t002:** EORTC C-30 and PR-25 scores at baseline, completion of SBRT, and at 12- and 24-months post-SBRT according to prostate volume (Repeated ANOVA analysis results).

	Prostate Volume ≤ 60 cm^3^					Prostate Volume > 60 cm^3^				
Item	Baseline	SBRT Completion	12 m	24 m	*p*-Value	Baseline	SBRT Completion	12 m	24 m	*p*-Value
EORTC QLQ-C30		Mean score ± SD								
Global health status/ Quality of life	61.2 ± 19.0	59.3 ± 17.9	62.8 ± 17.7	63.3 ± 17.5	0.0024	62.7 ± 17.8	56.5 ± 17.3	58.2 ± 20.2	61.35 ± 16.0	0.0126
Physical functioning	80.3 ± 18.4	80.4 ± 17.0	78.9 ± 17.2	75.4 ± 19.6	<0.0001	81.3 ± 16.8	81.7 ± 16.3	78.59 ± 19.1	76.21 ± 19.7	0.0056
Role functioning	88.8 ± 18.6	85.7 ± 19.7	86.8 ± 17.7	84.4 ± 20.4	0.0026	88.4 ± 18.2	86.8 ± 16.0	83.87 ± 22.9	83.02 ± 20.8	0.0766
Emotional functioning	71.3 ± 20.9	71.8 ± 21.7	73.2 ± 20.0	72.5 ± 21.1	0.3616	71.7 ± 19.2	71.9 ± 19.6	77.12 ± 16.9	71.52 ± 23.4	0.0144
Cognitive functioning	80.1 ± 20.4	79.9 ± 19.6	79.3 ± 20.6	76.5 ± 21.4	0.0059	79.5 ± 21.9	81.4 ± 20.9	80.3 ± 22.7	78.6 ± 24.2	0.6807
Social functioning	83.7 ± 20.3	81.3 ± 20.9	83.8 ± 19.6	82.8 ± 20.4	0.1324	82.8 ± 20.2	79.7 ± 21.1	79.3 ± 21.6	77.5 ± 25.0	0.1797
Fatigue	27.8 ± 20.9	29.5 ± 20.5	29.7 ± 19.5	31.1 ± 20.2	0.0240	27.2 ± 19.8	27.3 ± 18.6	29.9 ± 22.2	33.9 ± 18.6	0.0315
Nausea and vomiting	3.4 ± 9.1	4.2 ± 9.5	3.91 ± 9.4	3.5 ± 8.6	0.4362	4.01 ± 8.91	4.1 ± 10.1	4.5 ± 8.4	5.0 ± 9.4	0.8557
Pain	18.8 ± 21.9	19.7 ± 21.2	18.7 ± 20.2	17.6 ± 19.6	0.4316	18.3 ± 19.1	21.2 ± 24.1	20.5 ± 21.9	22.1 ± 25.1	0.5361
Dyspnea	15.1 ± 22.3	15.0 ± 23.8	17.9 ± 24.7	17.1 ± 25.5	0.0895	16.3 ± 24.2	12.5 ± 19.1	17.5 ± 22.3	20.2 ± 21.8	0.0171
Insomnia	32.5 ± 30.1	30.5 ± 30.4	31.8 ± 29.8	29.6 ± 27.5	0.2518	27.0 ± 28.0	32.9 ± 29.6	31.0 ± 29.4	32.2 ± 24.5	0.2048
Appetite loss	9.9 ± 19.5	10.5 ± 19.0	10.3 ± 16.7	9.6 ± 17.6	0.8284	8.51 ± 18.1	12.9 ± 22.5	12.6 ± 20.4	15.5 ± 22.6	0.0614
Constipation	16.5 ± 23.5	16.9 ± 24.9	19.2 ± 24.3	21.6 ± 27.5	0.0073	14.3 ± 23.1	17.4 ± 23.8	15.0 ± 18.4	19.5 ± 23.2	0.3435
Diarrhea	7.9 ± 16.0	11.2 ± 18.7	8.2 ± 16.3	10.2 ± 18.4	0.0074	8.4 ± 15.9	15.5 ± 22.0	13.7 ± 21.7	12.3 ± 20.7	0.0268
Financial difficulties	20.3 ± 26.0	18.9 ± 25.5	16.9 ± 25.7	15.4 ± 25.1	0.0058	16.3 ± 25.5	19.7 ± 29.4	21.0 ± 28.0	20.9 ± 27.3	0.2855
EORTC QLQ-PR25										
Urinary symptoms/problems	22.2 ± 16.3	32.0 ± 19.6	25.7 ± 17.7	25.5 ± 17.4	<0.0001	23.5 ± 17.3	37.3 ± 18.3	26.9 ± 18.1	28.1 ± 15.0	<0.0001
Incontinence aid	0.8 ± 7.3	4.0 ± 14.2	1.5 ± 6.9	4.3 ± 12.7	0.0661	0.18 ± 5.1	3.7 ± 16.5	1.6 ± 8.7	5.64 ± 10.2	0.5400
Bowel symptoms	8.31 ± 10.3	9.7 ± 11.2	10.3 ± 13.6	11 ± 12.4	0.0052	7.2 ± 7.6	12.3 ± 10.6	11.5 ± 12.7	12.3 ± 12.8	<0.0001
Treatment-related symptoms	15.5 ± 14.6	16.2 ± 13.6	18.6 ± 16.8	19.5 ± 16.1	<0.0001	13.6 ± 13.6	14.5 ± 12.1	17.2 ± 15.1	20.0 ± 13.9	<0.0001
Sexual activity	65.4 ± 22.6	69.6 ± 23.1	68.6 ± 22.4	67.9 ± 23.1	0.0099	63.2 ± 23.4	70.2 ± 21.2	68.0 ± 23.3	70.8 ± 21.8	0.0086
Sexual functioning	62.4 ± 23.8	63.8 ± 22.6	58.5 ± 24.5	53.1 ± 25.8	0.0005	57.9 ± 22.2	65.0 ± 19.7	54.0 ± 27.3	49.1 ± 26.5	0.0538

Abbreviations: EORTC QLQ-C30 = European Organization for Research and Treatment of Cancer core quality-of-life questionnaire; EORTC QLQ-PR25 = EORTC prostate cancer module; SD, standard deviation; m, months.

**Table 3 jpm-13-00233-t003:** Changes in health-related quality of life scores from baseline to month 24 according to prostate volume.

Item	Prostate Volume ≤ 60 cm^3^	Prostate Volume > 60 cm^3^	*p*-Value
	n = 415	n = 115	
EORTC QLQ-C30	Mean (±SD) change from baseline to 24 months		
Global health status	−2.18 ± 18.31	1.37 ± 16.93	0.0621
Functional scales			
Physical functioning	4.89 ± 19.01	5.18 ± 18.3	0.8840
Role functioning	4.39 ± 19.53	5.45 ± 19.54	0.6068
Emotional functioning	−1.18 ± 21.07	0.26 ± 21.32	0.5180
Cognitive functioning	3.67 ± 20.96	0.9 ± 23.07	0.2206
Social functioning	0.86 ± 20.38	5.27 ± 22.67	0.0457
Symptom scales			
Fatigue	−3.35 ± 20.61	−6.67 ± 19.23	0.1216
Nausea and vomiting	−0.06 ± 8.86	−1.01 ± 9.19	0.3133
Pain	1.27 ± 20.81	−3.81 ± 22.16	0.0228
Dyspnoea	−1.96 ± 23.95	−3.82 ± 23.04	0.4578
Insomnia	2.87 ± 28.86	−5.18 ± 26.31	0.0072
Appetite loss	0.3 ± 8.59	−7.01 ± 20.39	0.0001
Constipation	−5.15 ± 25.55	−5.26 ± 23.17	0.9668
Diarrhoea	−2.38 ± 17.27	−3.83 ± 18.36	0.4324
Financial difficulties	4.94 ± 25.96	−4.64 ± 26.45	0.0005
EORTC-QLQ-PR25			
Symptom scales			
Urinary symptoms	−3.3 ± 16.92	−4.61 ± 16.65	0.4613
Incontinence aid	−3.56 ± 10.03	−5.46 ± 7.7	0.0603
Bowel symptoms	−2.69 ± 11.4	−5.02 ± 10.21	0.0480
Hormone treatment-related symptoms	−3.94 ± 15.39	−6.39 ± 13.81	0.1233
Functional scales			
Sexual activity	−2.52 ± 22.93	−7.57 ± 22.61	0.0365
Sexual functioning	9.3 ± 24.84	8.77 ± 24.42	0.8391

## Data Availability

The raw data are available in database of The GreaterPoland Cancer Center.

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
