# Peer review of "Large Prostate Volume Does Not Negatively Impact Health-Related Quality of Life in Patients with Prostate Cancer Treated with Ultrahypofractionated Stereotactic Body Radiotherapy"

_jpm, 2023, doi:10.3390/jpm13020233_

Round 1

Reviewer 1 Report

1.       For what reason did you devide the prostate volume at 60 cm3?

The number of patients with less than 60 cm3 is considerably larger than the number of patients with more than 60 cm3.

2.       Why do you think there is a difference in sexual function between patients with less than 60 cm3 and those with more than 60 cm3?

Author Response

First of all I would like to Thank the review of our paper on comparison the quality of life patients treated with SBRT.

       Regarding the first comments I would like to explain that the cut-off volume of 60 cc for prostate gland was chosen because of two main reasons:

The first in many papers which deals with treatment for prostate cancer with brachytherapy exactly this level of volume is threshold for less versus higher level of side effects related to treatment. So, because SBRT could be so kind of competition to brachytherapy regarding which method of therapy is more suitable for our patient information regarding this parameter could impact the choice of therapy methods.

In the currently recruiting patients randomized clinical trial (GU 005) comparing SBRT to IMRT according to the inclusion criteria patients into the study are accepted with the volume of prostate less than 60 cc.

The second reason is practical too, because prostate cancer in many cases coexisted with non-malignant enlarged prostate gland volume and the practical volume cut-off for surgeons (urologists) who operated using different methods for non-malignant (Benign-Prostate-Hyperplasia BPH)/Malignant gland is volume about 60 cc.

         Regarding the second comment: In general, higher volume of prostate gland could be an issue for radiation oncologist as well urologists. From radiotherapy point of view in case of larger volume of the target is it harder to protect the small volume of Organ at Risk as the Neurovascular Bundle (NVB) which is located very closely to the capsule of prostate. That’s one reason why radiotherapy for in such situation is more challenging.

Best regards,

Piotr Milecki

Reviewer 2 Report

The authors present a prospective study aimed at finding how an enlarged prostrate volume affects the quality of life of patients after fractionated radiotherapy. The study is well done, with low between-group variability. However, the manuscript can be better with more visual representation of the data and some additional analyses. There are also a few inconsistencies in the interpretation of the results that needs to be corrected:

1.      Can the authors provide more justification for why they chose 60 mm3 as the cut off point for prostrate volume? Will the study results have been different if the threshold was different?

2.      Why was a 10-point difference chosen as the cut-off for clinical relevance? The authors have cited one work in this regard, but is this a widely accepted threshold in the field?

3.      Line 162 – The authors have to consistent with what they consider to be clinically meaningful. Here, the difference in sexual activity between pre-op and 24 months is <10 points but is arbitrarily considered to be clinically meaningful. Further the differences in sexual activity between the low and high prostrate volume groups are also not high enough to be considered clinically meaningful according to the authors’ definitions.

4.      Can the authors also plot x-y graphs with prostrate volume on the x axis and different HRQOL metrics on the y axis and report on the correlation between HRQOL metrics at 24 months and prostrate volume? This will answer the question of whether prostrate volume is correlated with HRQOL metrics at 24 months.

5.      Discussions – Once the authors have reassessed what they consider to be clinically relevant according to the feedback above, please put more details in the discussions section about which exact metrics were affected by prostate volume.

Author Response

First of all I would like to Thank the review of our paper on comparison the quality of life patients treated with SBRT.

       Regarding the first comments I would like to explain that the cut-off volume of 60 cc for prostate gland was chosen because of two main reasons:

The first in many papers which deals with treatment for prostate cancer with brachytherapy exactly this level of volume is threshold for less versus higher level of side effects related to treatment. So, because SBRT could be so kind of competition to brachytherapy regarding which method of therapy is more suitable for our patient information regarding this parameter could impact the choice of therapy methods.

In the currently recruiting patients randomized clinical trial (GU 005) comparing SBRT to IMRT according to the inclusion criteria patients into the study are accepted with the volume of prostate less than 60 cc.

The second reason is practical too, because prostate cancer in many cases coexisted with non-malignant enlarged prostate gland volume and the practical volume cut-off for surgeons (urologists) who operated using different methods for non-malignant (Benign-Prostate-Hyperplasia BPH)/Malignant gland is volume about 60 cc.

It is very hard to predict the influence of prostate volume on the QOL without appropriate analysis. We made analysis for the volume as a continuous variable (not included in the manuscript) and we noted the trend showed correlation between negative impact of higher volume, however we underline the small number of patients in such. In general, we can conclude that higher volume (probably the most appropriate cut-off volume > 60 cc) has negative impact but maybe for some subgroup (lower or higher volume) the impact of volume could be modified by other factors.

In general, the EORTC QOL results interpretation for daily clinical practice is based on the manual provided by the EORTC.

According to the manual the raw QLQ-C30 scores can be transformed to scores ranging from 0 to 100. The use of these transformed scores has several advantages, but transformed scores may be difficult to interpret. For example, what does an emotional function score of 60 or a difference of 15 mean? Also, there are no grounds for regarding, say, an emotional function score of 60 as being equally good or bad as scores of 60 on the other functioning scales. However, there are a number of ways to ease the interpretation of QLQ-C30 results.

Current approaches to defining clinical thresholds for the EORTC QLQ-C30 vary substantially and each has its own strengths and weaknesses. The simplest and most straight-forward approach is to rely on the wording of the item or response categories themselves, and classify a patient as having a clinically important problem if s/he responds with at least “a little” for any given item or domain assessed on the 4 point response scale (i.e. “not at all,” “a little,” “quite a bit,” and “very much”). This approach can be problematic as it uses the same threshold for all QLQ-C30 domains. The QLQ-C30 items differ in item difficulty (i.e. they assess different levels of severity of a problem or symptom), suggesting that scores are not directly comparable across domains. For example, reporting “quite a bit” of vomiting probably indicates a different symptom level than reporting “quite a bit” of trouble with sleeping. So, currently we don’t have ideal toll for interpretation results for daily practice. The proposition cut-off of 10 points as a clinically relevance is the most used in the interpretation of results from different studies, but in addition it should be included the raw statistical results.

Best regards,

Piotr Milecki

Round 2

Reviewer 2 Report

I would like to thank the authors for their response. However, no changes have been made to the manuscript, even though I asked for major revisions in the first round of review. The authors provide their explanation on many of the concerns I raised but I do not understand why these explanations were not included in the revised manuscript. The authors should understand that I only asked for major revisions because I have significant concerns about the interpretation of the data. Hence, significant changes to the manuscript are absolutely needed before I can accept this.

1.      Thank you for explaining the reason why 60 mm3 was chosen as a cutoff. Please include this in either the introduction or discussion section. It is not at all obvious in the current manuscript why this cut off was chosen.

2.      As the authors noted in their response “We made analysis for the volume as a continuous variable (not included in the manuscript) and we noted the trend showed correlation between negative impact of higher volume, however we underline the small number of patients in such”. This is exactly what I asked for in point 4. If the authors have the data already analyzed, then why not include it in the manuscript? Also, the authors have 530 patients, which is hardly a small number of patients to draw statistical significance from.
This data, whether positive or negative, should be included in the manuscript, or at least provided as supplementary information and discussed in the main paper. It is not acceptable that the authors have already analyzed the data and not provided it, despite being asked for it explicitly.

3.      Thank you for explaining the subjectivity of the scoring system. I understand that these scores are hard to interpret but my concern was about the consistency of the analysis. If the scoring system was interpreted on a case-by-case basis, then absolute statements such as using a 10 point difference as “significant” should not be used. The explanation the authors provided should be integrated into the manuscript and used when interpreting the data.
To further elaborate, currently the manuscript shows the statistical significance for the different variables but the interpretation of what is clinically significant is done subjectively. The authors need to provide more details in the discussions section about how each variable that was considered to be “clinically significant” was deemed to be so. Currently it seems like the authors are arbitrarily selecting the variables they deem significant without any explanation. The subjectivity of this interpretation needs to be explained in detail, and arbitrary point differences should not be mentioned if they are not being used.

Author Response

First of all, I would like to apologize for the quality of my first round respond to the reviewer. The reasons for that were the fact that it was done in a hurry during my trip to USA. So, the statement that we made the analysis for the volume as a continuous variable was not valid, we had some plans only.

We have introduced some extra text in which it was explained why the 60 cc cut-off volume of prostate was proposed in the manuscript. We added some explanation regarding the issues of interpretation of raw data for analysis the QOL domains.

In addition, we provided the extra statistical correlation on the volume of prostate and the investigated domains of QOL.  In general, that analysis revealed no statistical correlation between the prostate gland volume and score level of QOL domains. So, we have the proposition that this analysis could be added as supplementary part of the manuscript or it will withdraw from the main body of the manuscript. In general, the final conclusion of the paper is that volume of prostate is not important factor in decision process when we have “on the table” SBRT as a possible method of treatment.

Once again sorry for some mistakes.

Best regards

Piotr Milecki

Round 3

Reviewer 2 Report

I would like to thank the authors for responding to my comments and providing the necessary explanations and figures. I would like to recommend you to please proofread the manuscript for grammatical errors before submitting the final version, especially for the newer edits.